SciPost Physics

Submission

# Pair binding and Hund's rule breaking in high-symmetry fullerenes

Roman Rausch, Christoph Karrasch

Technische Universität Braunschweig, Institut für Mathematische Physik,
Mendelssohnstraße 3, 38106 Braunschweig, Germany

May 28, 2025

## Abstract

Highly-symmetric molecules often exhibit degenerate tight-binding states at the Fermi edge. This typically results in a magnetic ground state if small interactions are introduced in accordance with Hund's rule. In some cases, Hund's rule may be broken, which signals pair binding and goes hand-in-hand with an attractive pair-binding energy.

We investigate pair binding and Hund's rule breaking for the Hubbard model on high-symmetry fullerenes $C_{20}$, $C_{28}$, $C_{40}$, and $C_{60}$ by using massive, large-scale density-matrix renormalization group calculations. We exploit the SU(2) spin symmetry, the U(1) charge symmetry, and optionally the Z(N) spatial rotation symmetry of the problem.

For $C_{20}$, our results agree well with available exact-diagonalization data, but our approach is numerically much cheaper. We find a Mott transition at $U_c \sim 2.2t$, which is much smaller than the previously reported value of $U_c \sim 4.1t$ that was extrapolated from a few datapoints. We compute the pair-binding energy for arbitrary values of $U$ and observe that it remains overall repulsive.

For larger fullerenes, we are not able to evaluate the pair binding energy with sufficient precision, but we can still investigate Hund's rule breaking. For $C_{28}$, we find that Hund's rule is fulfilled with a magnetic spin-2 ground state that transitions to a spin-1 state at $U_{c,1} \sim 5.4t$ before the eventual Mott transition to a spin singlet takes place at $U_{c,2} \sim 11.6t$. For $C_{40}$, Hund's rule is broken in the singlet ground state, but is restored if the system is doped with one electron. Hund's rule is also broken for $C_{60}$, and the doping with two or three electrons results in a minimum-spin state.

Our results support an electronic mechanism of superconductivity for $C_{60}$ lattices. We speculate that the high geometric frustration of small fullerenes is detrimental to pair binding.

## 1  Introduction

Hund's rule is a well-known effect from atomic physics: An electronic shell that is degenerate in the one-particle picture will in fact split up upon introducing interactions, and the lowest-energy state features a maximum spin [1] which reduces interelectronic repulsion [2]. Intuitively, same-spin electrons avoid getting close to each other by virtue of the Pauli principle, while opposite-spin electrons can occupy the same space. In nuclei, the interfermionic interactions are attractive as a result of the strong force and the opposite situation occurs: An even number of nucleons in a degenerate shell will always form a singlet state.

Hund's rule is mostly valid in molecules as well: If a free-electron model on some molecular geometry produces degenerate shells, the inclusion of weak interactions will typically lead to a maximum-spin ground state. For some molecules, however, Hund's rule may be broken, which signals effective attraction similar to nuclei [3]. This is a case of interest because molecules with attractive pair binding may serve as building blocks for molecular lattices with global superconductivity, with a purely electronic mechanism that can be understood from the individual molecules alone.

Pair binding can be quantified by evaluating the pair binding energy

$$E_b = E_0(N_{\text{tot}}) + E_0(N_{\text{tot}} + 2) - 2E_0(N_{\text{tot}} + 1), \tag{1}$$

where $E_0(N_{\text{tot}})$ is the state with the lowest overall energy of an isolated molecule in the sector with $N_{\text{tot}}$ electrons. If $E_b < 0$, then an ensemble of uncoupled molecules will prefer a state with doped electrons paired up on the same molecule, rather than being split on separate ones. Weak intermolecular hopping is then expected to render a molecular lattice superconducting.

It is known that a large shell degeneracy, which is generally related to a non-Abelian molecular symmetry, is a key ingredient for pair binding [3]. High-symmetry molecules can also more easily crystallize into lattices. The most interesting molecules are thus the highly symmetric ones, e.g., those forming Platonic or Archimedean solids. Of particular importance is the $C_{60}$ fullerene, which is an Archimedean solid (truncated icosahedron, symmetry group $I_h$). It is insulating at half-filling (i.e., it features a full shell, and thus a singlet ground state) and has a threefold degenerate lowest unoccupied molecular orbital (LUMO), which can be chemically doped (see Tab. 1). Experimentally, a face-centered cubic (fcc) lattice of electron-doped $C_{60}$ molecules shows superconductivity with a critical temperature of $T_c \approx 30 - 40\text{K}$ [4,5]. This is relatively high for a phononic mechanism, so that an electronic mechanism is also being debated [6]. In order to investigate the latter, one needs to solve a fermionic model given by the Hubbard Hamiltonian

$$H = -\sum_{ij\sigma} t_{ij} c_{i\sigma}^\dagger c_{j\sigma} + \text{h.c.} + U \sum_i n_{i\uparrow} n_{i\downarrow} \tag{2}$$

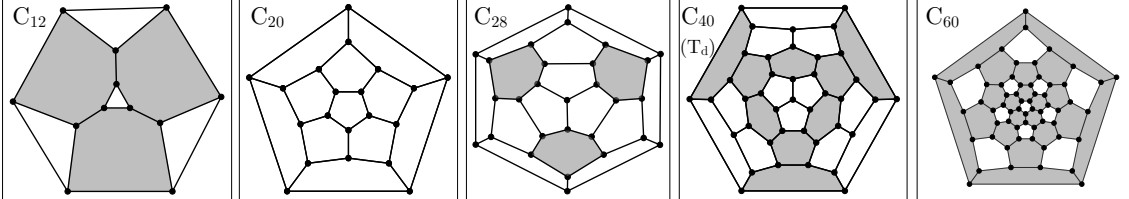

Figure 1: Schlegel diagrams (nearest-neighbor graphs) of the high-symmetry fullerenes studied in this work.

on the molecular geometry. Here, $t_{ij}$ is the hopping matrix, $c_{i\sigma}^\dagger$ ($c_{i\sigma}$) is the creation (annihilation) operator for an electron with spin $\sigma =\uparrow, \downarrow$ at site $i$, $U$ is the onsite Coulomb interaction, and $n_{i\sigma} = c_{i\sigma}^\dagger c_{i\sigma}$ is the electron density. One typically employs the tight-binding approximation where $t_{ij}$ takes the value $t$ between nearest neighbors, set to be the unit of energy $t = 1$ from now one.

Since the problem for $C_{60}$ with $L = 60$ sites is quite difficult to tackle theoretically, early focus has been on smaller molecules. The truncated tetrahedron with $L = 12$ sites can be regarded as a coarse-grained simplified version of $C_{60}$ (see Fig. 1). It is also insulating, has a threefold degenerate LUMO as well as a similar geometry, albeit with 3 triangles instead of 12 pentagons (group $T_d$), see Tab. 1. For this reason, it is sometimes called $C_{12}$, although it is not a proper fullerene. Using exact diagonalization (ED), White et al. demonstrated attractive pair binding with a peak of $E_b \sim -0.02$ around $U \sim 2$ within the Hubbard model [3]. Interestingly, a similar value is also observed for the cube ($L = 8$), which shows an additional peak of $E_b \sim -0.03$ at $U \sim 10$ [3]. It was found that if longer-ranged interactions are taken into account within the extended Hubbard model, pair binding shifts towards repulsion.

The smallest proper fullerene is $C_{20}$ (dodecahedron, group $I_h$). Pair binding on $C_{20}$ has been investigated using ED within the Hubbard model, but since large-scale ED calculations are quite expensive for $L = 20$, only $U = 2$ and $U = 5$ were studied, and the missing data was filled by Quantum Monte Carlo (QMC) [7]. The surprising result was a repulsive $E_b > 0$ in the whole $U$-range, and this was unchanged within the extended Hubbard model [8].

Attempts to treat $C_{60}$ yielded conflicting results. While extrapolations from perturbation theory to intermediate $U$ predicted a minimum-spin doped state [9, 10], QMC calculations yielded maximum-spin state [11]. Using the density-matrix renormalization group (DMRG), the related simpler $t - J$ model was investigated on $C_{60}$, and a minimum-spin state was found [12]. However, the $t - J$ model arises in the strong-$U$ limit, whereas carbon is governed by intermediate $U$.

In this paper, we want to address the following questions: i) Is Hund's rule fulfilled for the intermediate fullerenes between $C_{20}$ and $C_{60}$? Because of the high-symmetry requirement, this reduces the interesting candidates to just $C_{28}$ and $C_{40}$ (both of $T_d$ symmetry, see Fig. 1). ii) What can we learn about the full Hubbard model on $C_{60}$? For this problem, we give the best possible estimate using massive, large-scale DMRG computations that are significantly more advanced than anything that can be found in the literature. iii) What is the reason that some molecules exhibit Hund's rule breaking, while others do not?

## 2 Technical details

We investigate finite molecular geometries within the pure Hubbard model (2) by using the DMRG method, which approximates the wavefunction variationally within the class of matrix-product states (MPS) [13]. The DMRG exploits the property that physical ground states are only entangled locally (area law) and can thus be accurately represented by MPS. The key control parameter is the so-called bond dimension $\chi$, which is related to the number of variational parameters. The area law renders the application to one-dimensional systems particularly efficient, but all finite systems necessarily only have finite entanglement which can be captured by a sufficiently large $\chi$. Hence, the method can also yield very accurate results for molecules that are not too large [12,14–17]. For this reason, it is widely employed in quantum chemistry [18].

To gauge the accuracy of our DMRG calculations, we use the energy variance per site:

$$\mathrm{var}(E)/L = \big|\langle H^2 \rangle - \langle H \rangle^2 \big|/L, \tag{3}$$

which should vanish for an exact eigenstate and thus serves as a measure of how far one is from the exact solution. In cases where the exact energy is known, one typically finds that the variance is linearly related to the true error [19], and we will therefore use this as an extrapolation scheme for the energy. We employ the two-site DMRG algorithm to grow the bond dimension on the first iterations before switching to the cheaper one-site algorithm with fluctuations [20]. The DMRG requires a mapping of the sites to a one-dimensional chain, and the molecular geometry generates long-range hopping terms whose range we minimize by applying the reverse Cuthill-McKee algorithm to the graph $t_{ij}$ [21]. The maximum hopping range for our systems is shown in Tab. 1. The higher the value, the more complex the problem becomes for the DMRG. The most difficult case considered here is $C_{60}$ with a range of 10, which is roughly equivalent in complexity to a 10-leg ladder. Finally, one must find a compact matrix-product operator representation [22].

The binding energy in Eq. (1) is small compared to the hopping $t$, typically $O(10^{-2}t)$, but is itself a result of the difference of large numbers.[1] Thus, it is crucial that the computed energies should be as accurate as possible, preferably to at least 4 digits. In order to boost the performance of the DMRG, we exploit the U(1) charge symmetry as

---

[1] We note that $t \sim 2.8\,\mathrm{eV}$ for carbon, which places a pair-binding energy of $E_b \sim 10^{-2}t$ on the scale of room temperature (0.025 eV), which is quite large in terms of absolute numbers.

| molecule | sym. | $W$ | shell filling | degeneracy | $r$ |
|----------|------|-----|---------------|------------|-----|
| cube | $O_h$ | 6.0 | closed, $\Delta = 2.0$ | 3 (LUMO) | 4 |
| $C_{12}$ | $T_d$ | 5.0 | closed, $\Delta = 1.0$ | 3 (LUMO) | 5 |
| $C_{20}$ | $I_h$ | 5.236 | 2 | 4 (HOMO) | 6 |
| $C_{28}$ | $T_d$ | 5.414 | 4 | 4 (HOMO) | 7 |
| $C_{40}$ | $T_d$ | 5.596 | 2 | 3 (HOMO) | 9 |
| $C_{60}$ | $I_h$ | 5.618 | closed, $\Delta = 0.757$ | 3 (LUMO) | 10 |

Table 1: Symmetry and bandstructure properties ($U = 0$; tight-binding or Hückel limit) of high-symmetry molecules, obtained by diagonalizing the hopping matrix $t_{ij}$. $W$ denotes the bandwidth. i.e., the difference between highest and lowest eigenenergy. The shell filling is the number of electrons in the HOMO for open shells, and $\Delta$ is the HOMO-LUMO gap closed shells. The degeneracy refers to the LUMO (HOMO) for closed (open) shells. The hopping range $r$ is the graph bandwidth of $t_{ij}$ after applying the reverse Cuthill McKee algorithm, which influences the complexity of the problem for the DMRG.

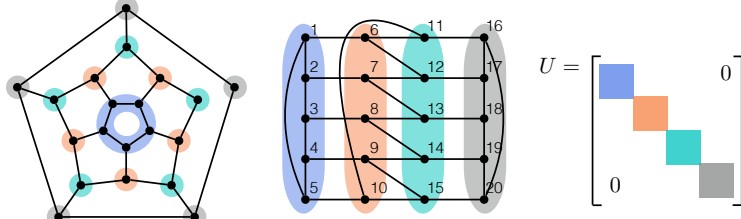

Figure 2: Left: Schlegel diagram of the $C_{20}$ hopping matrix with the points connected by Z(5) rotations shaded in the same color. Middle: The same graph, but deformed to a ladder-like structure with the points connected by rotations placed on the rungs. The depicted enumeration of the sites leads to a maximal hopping range of 6 (see Tab. 3). Right: The unitary transformation that rotates the molecule is block-diagonal.

well as the SU(2) spin symmetry [23]. This approach also allows us to directly compute the ground state energy $E_0(N_{\text{tot}}, S_{\text{tot}})$ in the different charge and spin sectors labelled by $N_{\text{tot}}$ and $S_{\text{tot}}$, respectively. Half filling corresponds to $N_{\text{tot}} = L$. We optionally exploit the Z(N) rotational symmetry (see Fig. 2), which is exactly analogous to the case of the transverse momentum for cylinders [24–26] and is briefly explained in the next section. We note that the exploitation of the SU(2) spin symmetry has the effect that the bond dimension $\chi_{\text{SU(2)}}$ used in the numerics corresponds to a larger effective bond dimension if only spin-U(1) is exploited, i.e., $\chi = g\chi_{\text{SU(2)}}$, with the gain factor of $g = 3 \pm 0.2$ in our case. It also allows us to directly target the total spin quantum number $S_{\text{tot}}$ instead of merely its projection $S_{\text{tot}}^z$.

## 2.1   Z(N) symmetry

In this section, we explain how to exploit the Z(N) molecular symmetry and discuss the corresponding numerical gain.

The $L \times L$ hopping matrix $\underline{t}$ is invariant with respect to permutations $\underline{P}$ of the orbitals that correspond to rotations of the molecule by a discrete angle (see Fig. 2). This permutation matrix defines a unitary transformation $\underline{U} = \underline{P} = \exp(\frac{2\pi i}{N}\underline{J})$, where $\underline{J} = \underline{J}^\dagger$ is hermitian. For an $N$-fold rotation, at most $N$ sites transform into each other, so that $\underline{U}$ is block-diagonal. The invariance under the permutation implies $[\underline{t}, \underline{U}] = 0$ and we can find common eigenvectors of $\underline{t}$ and $\underline{U}$. Because $\underline{U}$ is unitary with $\underline{U}^N = 1$, its eigenvalues are given by $\lambda_J = \exp\left(\frac{2\pi i}{N}J\right)$, $J = 0, 1, \ldots N - 1$, so that $J$ can be taken as an additional quantum number of the transformed orbitals and has the physical meaning of a discrete angular momentum.

This unitary transformation can be carried over to the many-body regime by letting it act on the creation and annihilation operators, $c_{\mathbf{k}\sigma}^\dagger = \sum_i U_{i\mathbf{k}}c_{i\sigma}^\dagger$ and $c_{i\sigma}^\dagger = \sum_{\mathbf{k}} U_{i\mathbf{k}}^* c_{\mathbf{k}\sigma}^\dagger$, where we use the bold letters to indicate the transformed orbitals. Application to the Hubbard term of the Hamiltonian (2) yields

$$\begin{aligned}
H_{\text{Hubbard}} &= U \sum_i c_{i\uparrow}^\dagger c_{i\uparrow} c_{i\downarrow}^\dagger c_{i\downarrow} \\
&= U \sum_i \sum_{\mathbf{klmn}} U_{i\mathbf{k}}^* U_{i\mathbf{l}} U_{i\mathbf{m}}^* U_{i\mathbf{n}} c_{\mathbf{k}\uparrow}^\dagger c_{\mathbf{l}\uparrow} c_{\mathbf{m}\downarrow}^\dagger c_{\mathbf{n}\downarrow},
\end{aligned} \tag{4}$$

where some terms vanish by angular momentum conservation. The newly-generated terms can be classified according to the number of distinct indices in the tuple ($\mathbf{klmn}$) as local, 2-site, 3-site, and 4-site terms. The interaction is hence no longer strictly local, but the non-locality is limited to segments of at most $N$ sites and the Hamiltonian can still be efficiently

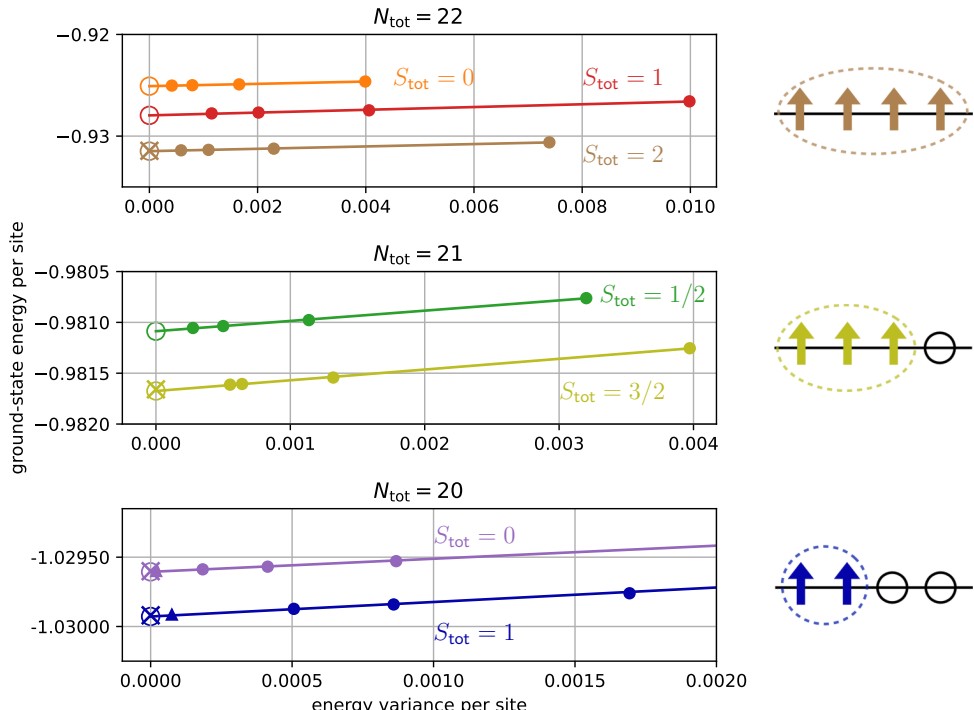

Figure 3: DMRG data for the ground-state energy per site for $C_{20}$ at $U = 2$ in various sectors of the particle number $N_{tot}$ and the total spin $S_{tot}$. Bullets: SU(2)×U(1)-symmetric calculation, bond dimensions $\chi_{SU(2)} \leq 10\,000$. Triangles: SU(2)×U(1)×Z(5)-symmetric calculation, $\chi_{SU(2)} = 15\,000$ (one datapoint in selected sectors). The DMRG values are linearly extrapolated in the variance per site (which decreases with increasing bond dimension), and the hollow circles indicate the extrapolated results. The crosses show the reference values from exact diagonalization. The right-hand side schematically visualizes the filling of the HOMO (arrows: electrons, black circles: empty sites). Hund's rule holds true at any filling.

represented as a matrix-product operator. If one wants to account for the SU(2) spin symmetry, the terms must be further sorted into spin-conserving groups and represented by reduced matrix elements, which is documented in Ref. [27].

The advantage of exploiting the Z(N) symmetry is that the quantum number blocks in the MPS representation shrink roughly by a factor of $N$, so that one can work with larger bond dimensions. A downside is the need to perform the calculations for all possible values of $J$, which is, however, easily parallelizable. Another disadvantage is the need to switch from real to complex numerics. The fundamental question, however, is whether the entanglement increase due to the additional nonlocal terms will be fully compensated by the increased bond dimension. For this reason, the benefit of using Z(N) symmetry is *a priori* not clear and depends on the geometry of the system. For the fullerenes, we find that exploiting Z(N) is advantageous for $C_{20}$, $C_{28}$, and $C_{40}$ but not for the most interesting case of $C_{60}$.

# 3   Molecule $C_{20}$

The properties of the (extended) Hubbard model on $C_{20}$ were investigated in detail using large-scale ED both for the undistorted geometry ($I_h$ symmetry) and including Jahn-Teller

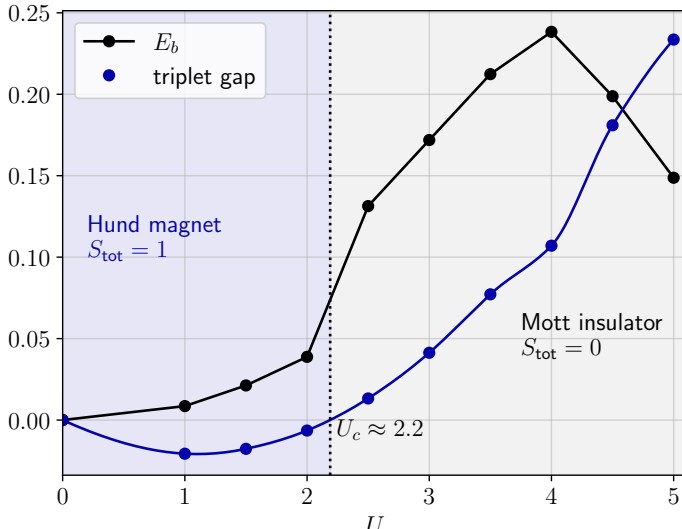

Figure 4: Triplet gap Eq. (5) and pair-binding energy Eq. (1) for $C_{20}$ as a function of $U$ at half filling $N_{\text{tot}} = 20$. $U_c \approx 2.2$ marks the metal-insulator (Mott) transition. Bullets indicate the raw data, lines are a result of Akima spline interpolation (triplet gap) or linear interpolation ($E_b$).

distortion (reduction to $D_{3d}$ symmetry) [7, 8]. Despite the exploitation of both the spin-U(1) and some spatial symmetries, the computational requirements are still quite heavy, with the Hilbert-space size reaching $3.4 \cdot 10^9$ in the undistorted case. Thus, only ED values for $U = 2$ and $U = 5$ are reported, and the missing values were filled in by QMC.

The shell structure of $C_{20}$ is metallic, as the HOMO is only partially occupied by two electrons (see Tab. 1). This allows one to study Hund's rule in the half-filled ground state, with the central question being whether the two shell electrons pair to a singlet or triplet. If $U$ is increased, local-moment formation will eventually set in, which results in a singlet ground state that is not due to Hund's rule breaking but rather attributed to a Mott transition. For the critical value, $U_c \sim 4.1$ was estimated based on a linear extrapolation of energies between $U = 2$ and $U = 5$ [7].

In Fig. 3, we benchmark our DMRG results for the ground state energy $E_0$ at $U = 2$ in various sectors of the particle number $N_{\text{tot}}$ and the total spin $S_{\text{tot}}$ against ED data. Hund's rule is fulfilled at any filling, i.e., the overall ground state features a maximum $S_{\text{tot}}$. We find that the DMRG is precise, with a typical absolute deviation in energy of the order of $10^{-4}$. For example, for $N_{\text{tot}} = 20$ and $S_{\text{tot}} = 0$, the exact result is $E_0 = -20.59202$, whereas the DMRG value obtained from an extrapolation in the energy variance per site reads $E_0 = -20.59211$. The DMRG result is accurate enough to clearly resolve the energy gaps between the sectors. For example, for the triplet gap

$$\Delta_1 = E_0 \left( N_{\text{tot}}, S_{\text{tot}} = 1 \right) - E_0 \left( N_{\text{tot}}, S_{\text{tot}} = 0 \right) \tag{5}$$

at half filling $N_{\text{tot}} = 20$, ED yields $\Delta_1 = -0.00636$, while the DMRG (extrapolated) value is given by $\Delta_1 = -0.00644$. This needs to be compared to the QMC data where the deviations seem to be at best of order $10^{-3}$. Furthermore, QMC suffers from the artifact that the energy depends on the magnetic quantum number for a given $S_{\text{tot}}$, which is strictly not the case within our SU(2)-invariant DMRG approach.

Since the DMRG approach is computationally cheap in comparision to ED, we can readily study arbitrary interactions. Our results are not consistent with the published QMC data in the intermediate-$U$ range. In Figure 4, we plot the triplet gap as a function

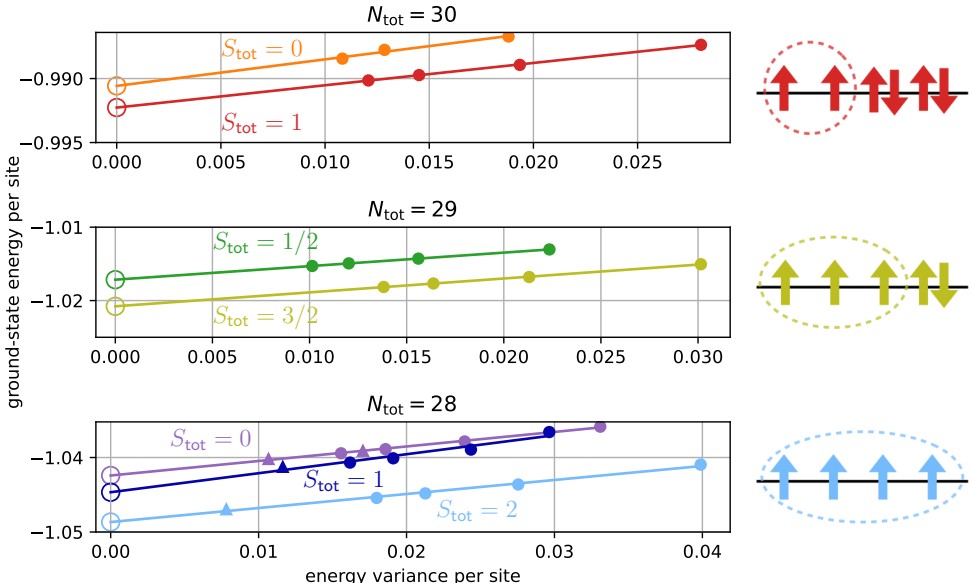

Figure 5: Ground-state energy per site and HOMO filling for $C_{28}$ at $U = 2$. The bond dimensions read $\chi_{\mathrm{SU}(2)} \leq 10\,000$ for the SU(2)×U(1)-symmetric calculation (bullets) and $\chi_{\mathrm{SU}(2)} = 7000, 15\,000$ for the SU(2)×U(1)×Z(3)-symmetric calculation (triangles).

of $U$. Using spline interpolation, we place the transition from a metallic Hund magnet to a Mott insulator at $U_c \approx 2.2$, which is significantly smaller than the previously-reported $U_c \sim 4.1$. The smallness of this value entails that one needs to exercise caution when extrapolating perturbation theory results to intermediate couplings. We also find that the pair binding energy $E_b$ increases sharply in the region of $U = 3 - 4$ and in fact reaches a local maximum. Nevertheless, we always find $E_b > 0$, i.e., repulsive pair binding in the whole $U$-region.

## 4  Molecule $C_{28}$

$C_{28}$ has tetrahedral symmetry and is metallic like $C_{20}$. It features a fourfold degenerate HOMO, which is partially occupied by four electrons in the ground state [28–32]. The fourfold degeneracy is a result of an accidental degeneracy of the $T_2$ and $A_1$ levels [29]. Due to its protruding tetrahedral bonds, it has been predicted that $C_{28}$ will form a diamond lattice [33–35], but this has not yet been realized chemically.

Figure 5 shows raw DMRG data as well as extrapolated values for the ground-state energy at $U = 2$ in different sectors of $N_{\mathrm{tot}}$ and $S_{\mathrm{tot}}$. We clearly see that Hund's rule holds true for all fillings, and the half-filled ground state ($N_{\mathrm{tot}} = 28$) is a quintet with $S_{\mathrm{tot}} = 2$. Tackling the Hubbard model on $C_{28}$ is considerably more costly than in the case of $C_{20}$, and the smallest energy variance per site Eq. (3) that can be accessed numerically is larger by one order of magnitude. Consequently, we are unable to precisely evaluate the pair binding energy $E_b$. A tentative estimate yields $E_b = +0.02 \pm 0.005$ at $U = 2$.

Figure 6 (a) follows the triplet gap Eq. (5) as well as the quintet gap

$$\Delta_2 = E_0(N_{\mathrm{tot}}, S_{\mathrm{tot}} = 2) - E_0(N_{\mathrm{tot}}, S_{\mathrm{tot}} = 0) \tag{6}$$

as a function of $U$ at half filling. We see that magnetism is more robust than in the case of $C_{20}$, and a ground state with $S_{\mathrm{tot}} = 2$ persists up to $U_{c,1} \approx 5.4$, after which it becomes a triplet with $S_{\mathrm{tot}} = 1$. A transition to a Mott insulator eventually occurs at $U_{c,2} \approx 11.6$.

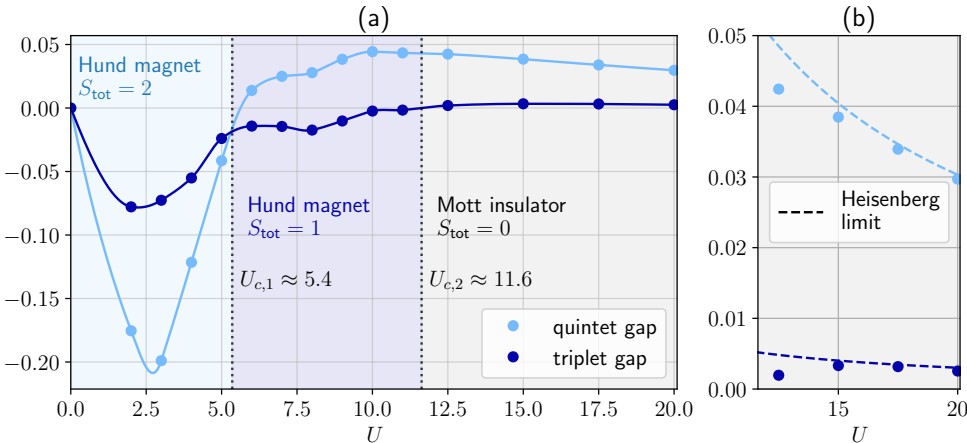

Figure 6: (a) The quintet gap Eq. (6) as well as the triplet gap Eq. (5) for $C_{28}$ as a function of $U$ at half filling $N_{\text{tot}} = 28$ (bullets: data, lines: Akima spline interpolation). (b) Zoom-in on the Mott-insulator region. The dashed lines show the Heisenberg limit $(U \gg 1)$ result: $\Delta_i = c_i/U$ with $c_2 = 0.15159$ and $c_1 = 0.01508$ [36].

Fig. 6 (b) compares the gaps in the Mott insulating phase with those obtained from a strong-coupling Heisenberg model calculation, which allows for accurate numerics [36]. We see that the DMRG treatment of the full fermionic problem is in full agreement with the effective Heisenberg model data.

# 5 Molecule $C_{40}$

$C_{40}$ has several isomers. The most stable isomer exhibits only a $D_{5d}$ symmetry [37–39] and at most twofold degenerate irreducible representations. For pair binding, we are interested in the more symmetric $T_d$ isomer (see Fig. 1) whose geometry is similar to $C_{28}$; it is also metallic but has a threefold degenerate HOMO without an accidental degeneracy.

Since tackling $C_{40}$ is numerically costly, we restrict ourselves to $U = 2$, which is a realistic value for carbon and also the point where thef pair binding energy peaks for $C_{12}$ [3]. Figure 7 shows raw DMRG data as well as the extrapolated value for the ground-state energy at various $N_{\text{tot}}$ and $S_{\text{tot}}$. The smallest site variance that can be accessed in our calculation is a factor of two larger than in the case of $C_{28}$ due to the increased system size and hopping range.

We find that $C_{40}$ exhibits a singlet ground state at half filling with a small gap to the triplet state, indicating Hund's rule breaking. When doped with one electron, however, we find that the maximum-spin sector $S_{\text{tot}} = 3/2$ has a slightly lower energy than the one with $S_{\text{tot}} = 1/2$. For a doping with two electrons, we are unable to resolve the energies between the sectors $S_{\text{tot}} = 0$ and $S_{\text{tot}} = 1$. This indicates that there is a tighter competition between the spin sectors for this geometry.

# 6 Molecule $C_{60}$

The half-filled ground state of $C_{60}$ is insulating with a completely filled shell, which is analoguous to $C_{12}$ (see Tab. 1). There are thus two relevant energy scales, the band gap $\Delta = 0.757$ and the bandwidth $W = 5.618$. We can discern the following regimes: (i) ultraweak coupling $U < \Delta$, (ii) weak coupling $\Delta < U < W$, (iii) intermediate coupling

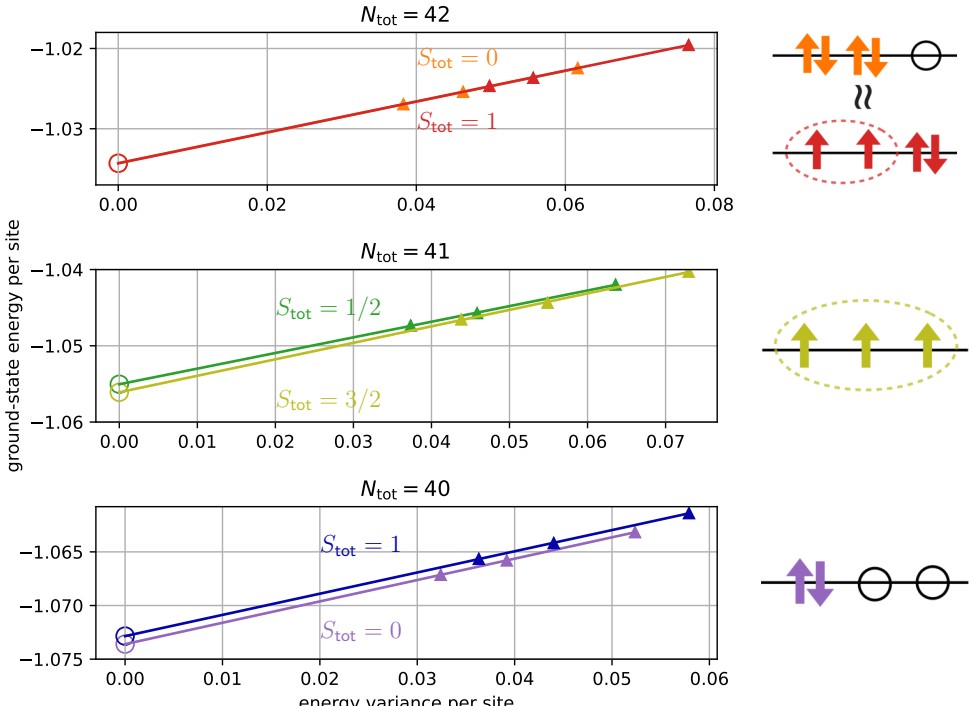

Figure 7: Ground-state energy per site and HOMO filling for $C_{40}$ ($T_d$) at $U = 2$. Only data for a SU(2)×U(1)×Z(3)-symmetric calculation is shown (triangles), and bond dimensions of up to $\chi_{\mathrm{SU}(2)} \leq 20\,000$ are employed.

$U \sim W$, and (iv) strong coupling $U \gg W$. The half-filled ground state is without question a total singlet $S_{\mathrm{tot}} = 0$ because of the insulating shell structure, and therefore not of particular interest. In alkali fullerides such as $K_3C_{60}$ [4], there are three doped electrons per $C_{60}$, with renders the ground state with $N_{\mathrm{tot}} = 63$ electrons the most interesting case.

Soon after the advent of the density-matrix renormalization group, the method was applied to $C_{60}$ in momentum space (i.e., using a basis where $t_{ij}$ is diagonal), but the only small bond dimensions were employed, and the results are only reliable for ultraweak coupling $U \sim 0.001 - 0.01$ [40]. The ground state energy with doping was computed, but the total spin and the pair binding energy were not investigated. In the strong-coupling limit, the undoped Hubbard model simplifies to the Heisenberg model and has been solved with high accuracy using the DMRG [15] as well as via an approach based on neural networks [41]. The undoped strong-coupling limit is useful in understanding the effects of frustration due to the pentagon faces (see App. A), but it is distinct from the doped weak-coupling regime investigated here.

There has been a number of studies in the doped weak-coupling regime. Using QMC, a maximum-spin ground state was found [11]. A minimum-spin ground state was predicted from extrapolating perturbation theory in $U$ [9, 10] and also for the $t - J$ model with a large $J = 0.2 - 1$ [12] using the DMRG with U(1) symmetry. The case $J \to 1$ is assumed to mimic the intermediate-coupling regime with $J = 4t^2/U$ and hence $U/t \sim 4$. However, correction terms such as three-site terms [42] or $O(t^4/U^3)$ contributions [43] have not been considered in this work. Without these corrections, the $t - J$ model should be viewed as similar to, but distinct from the Hubbard model.

$C_{60}$ poses a very tough problem: Since the system size grows by another 50% compared to $C_{40}$ and the hopping range increases further, we find that the best achievable variance per site within the DMRG deteriorates further by one order of magnitude. Furthermore,

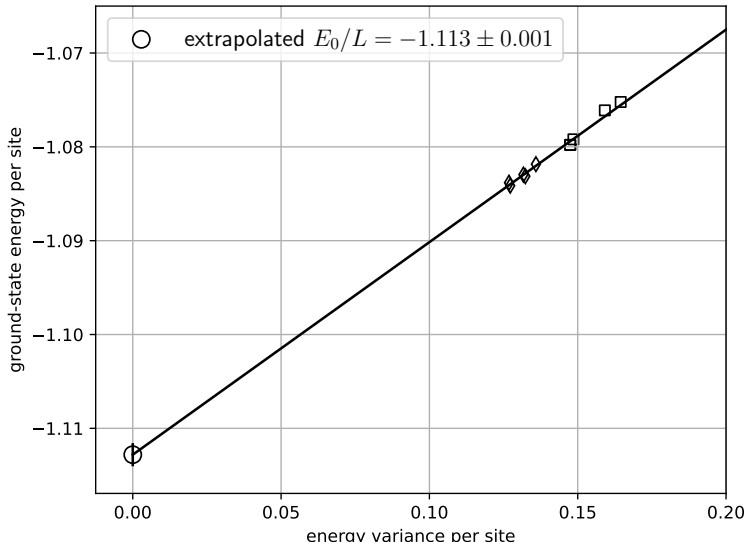

Figure 8: Ground-state energy per site for $C_{60}$ at half-filling $N_{tot} = 60$ and a total spin $S_{tot} = 0$ at $U = 2$. Only data for a SU(2)×U(1)-symmetric calculation is shown. The bond dimensions are in the range $14\,000 \leq \chi_{SU(2)} \leq 16\,000$ (diamonds) and $10\,000 \leq \chi_{SU(2)} < 14\,000$ (squares). The energy is extrapolated from the five most accurate values (diamonds), while the error bar is estimated from the variation when considering all ten points (diamonds and squares); see the text for more details.

we observe that the energy variance is larger if the Z(5) symmetry is exploited, so that we restrict ourselves to SU(2)×U(1)-symmetric calculations. The increased hopping range also heightens the risk of getting stuck in a local minimum. We therefore use the following protocol: First, we perform 12 half-sweeps using the two-site algorithm with a bond dimension of $\chi_{SU(2)} = 2500$. We then increase the bond dimension as follows: $2500 \rightarrow 5000 \rightarrow 6000 \rightarrow 7000 \rightarrow 8000 \rightarrow 9000 \rightarrow 10\,000 \rightarrow 12\,000 \rightarrow 14\,000 \rightarrow 16\,000$ and perform 4 half-sweeps at each bond dimension using the one-site algorithm with perturbations. We carry out additional calculations where the bond dimension increases more aggressively, namely $3000 \rightarrow 6000 \rightarrow 9000 \rightarrow 12\,000 \rightarrow 15\,000$ as well as $4000 \rightarrow 8000 \rightarrow 12\,000 \rightarrow 16\,000$, and finally $5000 \rightarrow 10\,000 \rightarrow 15\,000$. This yields ten data points with $\chi_{SU(2)} \geq 10\,000$ and five data points with $\chi_{SU(2)} \geq 14\,000$. We extrapolate the energy using both of these two sets and take the difference between the two extrapolated values as an error estimate. The result is shown in Fig. 8 for the half-filled ground state and in Fig. 9 for finite doping. We provide a summary of the results in Tab. 2 for a potential benchmarking with other methods.

From Fig. 9 one sees that a minimum-spin state is clearly below a maximum-spin state for the 2-electron and 3-electron doped case. This is not only true for the extrapolated values, but also at each bond dimension $\chi_{SU(2)}$. For a doping with 4 electrons, we were not able to resolve the energies (similar to the case of $C_{40}$, see Fig. 7). Overall, our result is consistent with both the perturbation theory prediction and the data obtained within the $t - J$ model and supports an electronic mechanism for superconductivity in $C_{60}$ lattices.

## 7 Discussion

We have investigated Hund's rule breaking in high-symmetry fullerenes $C_{20}$, $C_{28}$, $C_{40}$ ($T_d$), and $C_{60}$ using large-scale DMRG calculations for the Hubbard model.

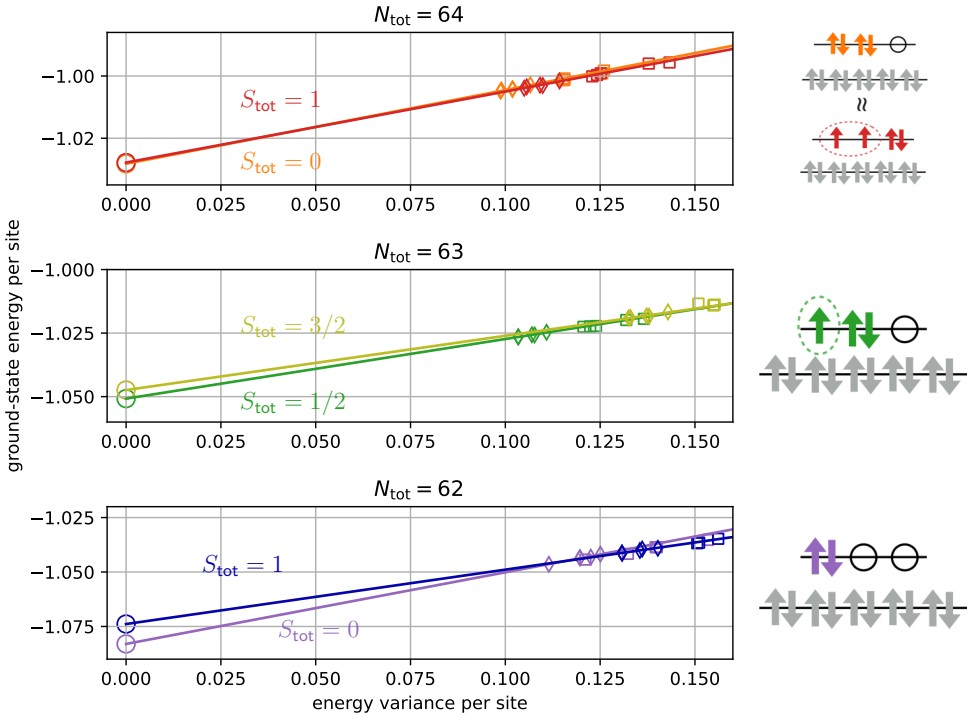

Figure 9: Ground-state energy per site for $C_{60}$ at $U = 2$. Symbols and bond dimensions are as in Fig. 8. The right-hand side schematically visualizes the filling of the HOMO (colored arrows: active electrons, grey arrows: inactive electrons of the full shell, black circles: empty sites).

For $C_{20}$, we find good agreement with previously-published ED data for $U = 2$ and $U = 5$, but the DMRG approach is numerically cheap and allows us to compute the pair-binding energy as a function of arbitrary $U$. We find that $C_{20}$ remains repulsive and adheres to Hund's rule in the weak-coupling limit, and we estimate a critical value of $U_c \sim 2.2$ for the metal-insulator transition. The binding energy is repulsive in the whole $U$-range.

We observe that $C_{28}$ is also adhering to Hund's rule at weak coupling. The half-filled ground state is magnetic with a quintet-triplet transition taking place at $U_{c,1} \sim 5.4$ before a transition to a Mott singlet takes place at $U_{c,2} \sim 11.6$. The typical $U$-values for carbon $U \sim 2 - 5$ [9, 44, 45] place $C_{28}$ into the quintet Hund magnet phase. A lattice of $C_{28}$ molecules is therefore expected to feature spin-2 sites as the low-energy building blocks. An open question is about the robustness of the spin-2 state towards Jahn-Teller deformations or stabilizer atoms [46].

For $C_{40}$, the DMRG computation is more expensive and less accurate, and we have only considered $U = 2$. Our result is somewhat ambiguous: we find a singlet for the undoped ground state that breakes Hund's rule but a maximum-spin state when doped with one electron. We were unable to resolve the energies in the different sectors with two doped electrons.

For $C_{60}$, we document DMRG data that results from massive calculations which are significantly more advanced than anything that can be found in the literature. Our best DMRG estimate shows Hund's rule breaking at $U = 2$ for the doping with two and three electrons. We were unable to resolve the energies in the sectors with four doped electrons.

Finally, we address the question whether Hund's rule breaking can be connected with a feature in the fullerene geometry. We observe that this breaking becomes more likely with increased fullerene size. The salient geometric feature that also changes with fullerene size

| $N_{\text{tot}}$ | $S_{\text{tot}}$ | $E_0(N_{\text{tot}}, S_{\text{tot}})/L$ |
|---|---|---|
| 60 | 0 | $-1.113 \pm 0.001$ |
| 62 | 0 | $-1.083 \pm 0.008$ |
| 62 | 1 | $-1.074 \pm 0.003$ |
| 63 | 1/2 | $-1.051 \pm 0.001$ |
| 63 | 3/2 | $-1.047 \pm 0.004$ |
| 64 | 0 | $-1.0282 \pm 0.0014$ |
| 64 | 1 | $-1.0278 \pm 0.0005$ |

Table 2: Ground-state energy per site in various sectors of the particle number $N_{\text{tot}}$ and the total spin $S_{\text{tot}}$ for $C_{60}$ at $U = 2$ estimated using a SU(2)×U(1)-symmetric DMRG calculation. See the text for details about the extrapolation procedure.

is the increase of the number of hexagon faces. Since all fullerenes have exactly 12 pentagon faces, this goes hand-in-hand with a spatial separation of the pentagons (see Fig. 1). Only the pentagons are geometrically frustrated, so that geometric frustration is reduced with the fullerene size, and $C_n$ approaches the unfrustrated hexagonal lattice for large $n$. The reduced frustration also becomes apparent when studying the spin-only Heisenberg model on the same geometries, which we discuss in App. A. We hence speculate that geometric frustration in fullerenes is detrimental to pair binding and may be the reason why smaller fullerenes are repulsive. In this picture, $C_{40}$ is positioned at the crossover point between repulsive and attractive pair binding, whereas $C_{60}$ is on the attractive side. This is also consistent with the fact that the bipartite and unfrustrated cube shows attractive pair binding [3].

We hope that these results can help guide the selection of molecular building blocks in engineered electron systems, such as metal-organic frameworks [47, 48].

The difficulty of solving the Hubbard model on the $C_{60}$ geometry leads us to propose this system as a benchmark for quantum computing. Previous benchmark systems for quantum simulation were amenable to highly efficient classical tensor-network treatments [49, 50]. But if one flips the perspective and looks at the situation from the tensor-network (MPS) point of view, then $C_{60}$ presents an interesting and hard problem for the classical MPS algorithm, despite being a finite and moderately-sized system. Our paper gives benchmark values using the best possible traditional MPS approach with symmetry exploitation. It would also be interesting if this classical result can be further improved using methods like fermionic orbital optimization [51] or novel techniques based on neural networks [41].

Finally, the computational bottleneck in our approach shifts from ground-state determination to variance evaluation via Eq. (3), which involves calculating $H^2$ for a long-range Hamiltonian. Our systems can thus also serve as a testing ground for algorithms that aim to provide cheaper variance estimates [52].

## Acknowledgements

Discussions with Matthias Peschke are gratefully acknowledged.

| molecule | g.s. deg. | singlet gap | triplet gap | reference | frustration |
|----------|-----------|-------------|-------------|-----------|-------------|
| $C_{12}$ | 1 | 0.896 | **0.688** | [53] | weak |
| $C_{20}$ | 1 | **0.316** | 0.514 | [54] | strong |
| $C_{28}$ | 2 | 0.0702 | **0.015** | [36] | strong |
| $C_{40}$ ($T_d$) | 2 | **0.04(2)** | 0.13(3) | this work | strong |
| $C_{60}$ | 1 | 0.691 | **0.356** | [15, 41] | weak |

Table 3: Properties of the molecules at strong coupling $U \gg 1$, Eq. (7): ground-state degeneracy, singlet gap $\Delta_0 = E_1(S_{\text{tot}} = 0) - E_0(S_{\text{tot}} = 0)$, triplet gap $\Delta_1 = E_0(S_{\text{tot}} = 1) - E_0(S_{\text{tot}} = 0)$. The smaller gap is shown in bold.

# A    Strong-coupling limit

For reasons of completeness, we summarize the properties of the fullerenes in the strong-coupling limit $U \gg 1$. Except for $C_{40}$, this is a compilation of results known from existing literature.

The low-energy behaviour in the strong-coupling limit is governed by the Heisenberg model

$$H = \sum_{ij} J_{ij} \mathbf{S}_i \cdot \mathbf{S}_j, \tag{7}$$

where $J_{ij}$ is the matrix of exchange interactions that has the same structure as $t_{ij}$, and $\mathbf{S}_i = (S_i^x, S_i^y, S_i^z)$ is the vector of spin operators. We set $J_{ij} = 1$ between nearest neighbours.

In the Heisenberg model, the effect of strong frustration manifests itself in low-lying singlet states below the first triplet, and sometimes also in a degenerate ground state. The former effect entails that low-lying excitations are unconventional and different from magnons (spinflips), while the latter implies symmetry breaking in the form of a valence-bond-solid-like state.

Table 3 shows a summary of the strong-coupling results. For $C_{40}$, which was not studied before, we calculate the low-lying states of Eq. (7) using a SU(2)-invariant DMRG algorithm with a bond dimension of $\chi_{\text{SU}(2)} = 2000$, which translates into an energy variance per site of $\sim 10^{-5}$, which is sufficient to estimate the gaps.

$C_{12}$ and $C_{60}$ are characterized by spinflip excitations, i.e., the smallest gap is the triplet gap; the gaps are rather large and the ground states are unique. $C_{12}$ and $C_{60}$ can be viewed as weakly frustrated: Their lattices are non-bipartite, but the triangle and pentagon plaquettes are not able to induce the abovementioned prototypical features of frustrated spin systems.

The lowest excitations for $C_{20}$ and $C_{40}$ are singlets. For $C_{28}$, they are a triplet, but the ground state for both $C_{28}$ and $C_{40}$ is twofold degenerate. In fact, the ordinary tetrahedron has a twofold degenerate ground state which features singlet coverings, and this property seems to be transferred to tetrahedral $C_{28}$ and $C_{40}$. We categorize $C_{20}$, $C_{28}$, and $C_{40}$ as strongly frustrated.

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
