# Peer review of "Pair binding and Hund's rule breaking in high-symmetry fullerenes"

_SciPost Physics Core_

## Round 1 · Referee Report · Anonymous (Referee 1) · 2025-7-16

Strengths

1- Large-scale DMRG study of the Hubbard model on fullerene geometries 2- Incorporating certain spatial symmetries in DMRG 3- Most detailed evidence to date for an electronic pairing mechanism in $C_{60}$

Weaknesses

1- Little motivation for the Hubbard model (is it actually a good description of the electronic structure?) 2- Little background information (e.g. structure of molecular orbitals) 3- Not enough detail on implementing the $Z_N$ symmetry in DMRG 4- Insufficient accuracy and questionable energy extrapolation for $C_{60}$

Report

In this paper, Rausch and Karrasch perform large-scale DMRG simulations of a single-orbital Hubbard model on various high-symmetry fullerene geometries. The motivation is to uncover an electronic mechanism of effective attraction between electrons in the unfilled HOMOs of fullerenes, which could explain superconductivity in alkali-fullerene materials such as $K_3C_{60}$. Such a mechanism has been proposed long ago (see e.g. Ref. [6]), but evidence for it has so far been limited to highly approximate methods, which are not reliable at realistic interaction strengths.

The authors use charge- and spin-symmetry-resolved DMRG to obtain ground-state energy estimates in different charge and total-spin sectors. These allow them to extract two signatures of pairing: 1- The pair binding energy, namely the difference between the energy cost of doping one molecule with two electrons vs. two molecules with one electron each. If this is negative, it is energetically favourable for dopant electrons to form pairs on some molecules, which may then condense to form the superconducting state. 2- Violations to Hund's rule (i.e., a ground state with less than highest possible spin in the unfilled HOMO) indicate an effective attractive rather than repulsive interaction between the electrons, which is conducive to pair formation. In addition to total spin and charge, the authors also use an innovative technique to impose a subset of rotational symmetries of the fullerene molecule on the wave functions, which leads to improved results on the smaller molecules they study, but curiously not on $C_{60}$.

DMRG is the most accurate on the smallest molecule considered, $C_{20}$. Here, the authors compute the pair binding energy and find that pair formation is unfavourable. Accordingly, Hund's rule is obeyed by the ground state. For larger molecules, the accuracy achieved by DMRG is not enough to resolve the pair binding energy, but there is clear evidence that Hund's rule still holds. By contrast, the authors claim that Hund's rule is violated for $C_{60}$ doped with 2 or 3 (and possibly also 4) electrons, which would underpin an electronic mechanism of superconductivity in $K_3C_{60}$.

I see two crucial issues with the last claim:

1- It is not clear whether a single-band, nearest-neighbour Hubbard model is an adequate description of fullerene electronic structure. One can make an analogy with graphene, where strongly-correlated electron effects (e.g. in twisted bilayer graphene) depend on long-range Coulomb interactions. Furthermore, given the curvature of the molecule, it is less clear how to isolate a single chemically active orbital per atom similar to the $p_z$ orbital in graphene. Therefore, it is unclear whether this Hubbard model is an improvement over considering smaller active spaces, so verifying the predicted electronic structure (e.g. the multiplicity and symmetries of the predicted active molecular orbitals) against experiment or ab-initio numerics would be important.

2- The quality of DMRG results for $C_{60}$ is far from ideal. (This is not a criticism per se, given how challenging the problem is.) The authors tackle this problem by variance extrapolation, a now-standard technique for variational numerical studies. However, the range of variances considered is quite narrow, so extrapolating to zero variance is a precarious process, typically with high error bars. The authors use non-standard error estimates, which (looking at the plotted data points) appear rather too optimistic. Even so, all pairs of extrapolated energies are equal to within the reported error, so their ordering (which determines whether Hund's rule is violated) is not settled. To properly evaluate the paper's conclusions, it is essential that the authors present their data with more standard and rigorous statistical analysis.

Given the ambition of the project, the technical advances to DMRG (especially on spatial symmetry), and the potential impact of the results, I think the paper would be a valuable addition to SciPost, provided the above issues can be satisfactorily addressed.

Requested changes

Main requests:

1- Is it known whether the predicted active space (i.e. HOMO, LUMO, and nearby shells) match experimental or ab-initio predictions? I mean multiplicities, point-group irreps, etc. This would be crucial to know in order to assess whether the Hubbard model is a good description of the fullerene molecules.

2- Table 1 is quite thin on information. Some useful additions would be: - the irrep formed by the relevant HOMO/LUMO in addition to the degeneracy - for the open shells, state how many electrons would fit in them in addition to how many they contain - is $r$ the Hamiltonian range in real space or in the hybrid real/momentum space introduced in Sec. 2.1? How does it change upon the basis transformation? - for $C_{60}$ at least, a figure detailing the layout of the active space (i.e. the energy, multiplicity and point-group irreps of all shells near the Fermi energy) would be desirable

3- Section 2.1: The implementation of the $Z_N$ symmetry is the most interesting technical aspect of the paper, but very little detail is provided. In particular: - How do you order the orbitals $c_{ik}$ for DMRG? Are the sites that make them up automatically sorted together by the real-space Cuthill-McKee algorithm, or do you need to label them manually? - How do you implement Eq. (4) and the hopping term? To someone who's not an expert in this technique, this looks like a daunting task. You should give a detailed discussion of how to obtain the Hamiltonian in MPO (?) form, at least in the simple charge-conserved case. - The first sentence of the last paragraph is quite confusing. Do you mean that the eigenvalue under the $Z_N$ rotation can be imposed as a symmetry quantum number, similar to charge?

4- Report the energies and variances of all DMRG runs plotted in Figures 3-9, either as tables in the appendix or as supplementary text files. These would be useful both for benchmarking and to independent assessment of the extrapolated results and conclusions.

5- Figures 3-9: include error bars (i.e. the standard intercept error of the linear fit) for all extrapolations, as they are essential to assess the confidence one should put into the results. Also make sure they're visible (in Fig. 8, the only figure that has it, it's easy to miss, because it's just a vertical line overlapping with the grid line, adding the usual serifs at the end would help). If the error bars are too small to see, state that in the caption explicitly.

6- Figures 3-9: The aspect ratio of the figures makes it look like there's almost no extrapolation, and it's also hard to see how accurately the points fit to the extrapolation lines. Consider laying out the panels horizontally, so that each of them are more square-like.

7- Figure 9: It's difficult to see how good the straight line fit is given all data points are crammed in one corner. Add extra panels/insets zooming in on the data points.

8- Table 2 (and error bars of Figure 9): - Show the standard intercept error of the linear fit instead of the custom error estimate. (If you want to use the separate data sets, show the full intercept+error pair for both.) - Comment on the fact that the extrapolated ground-state energies for 62 and 63 electrons are equal to within 1σ (62 electrons: Δ=0.009, σ=0.008,0.003, 63 electrons: Δ=0.004, σ=0.004,0.001). How much should we still trust the ordering of the energy estimates?

9- Do the conclusions from DMRG match predictions from simpler calculations (e.g. ED restricted to the HOMO/LUMO)?

10- Is the two-stage breakdown of Hund's rule in $C_{28}$ related to the accidental degeneracy of the $T_2$ and $A_1$ irreps? What would Hund's rule predict if the degeneracy were lifted?

Minor comments on the manuscript:

11- $Z(N)$ is nonstandard notation for the group in question. The typical mathematical notation is $\mathbb{Z}_N$, but given the geometrical meaning of the symmetry group, $C_N$ is even more appropriate. Switch to either one of these.

12- In the abstract and the introduction: "massive, large-scale" is a bit excessive, "large-scale" would be enough.

13- In the abstract: "For $C_{40}$, Hund's rule is broken in the singlet ground state" Do you mean "at half filling"?

14- In the first paragraph of the introduction: Is the discussion of the nuclear anti-Hund's rule really relevant? If so, provide reference(s).

15- Eq. (1): What is $N_\mathrm{tot}$? How should one set it?

16- Figure 1: what do the colours of the plaquettes mean? Also, show the numbering of the sites used for DMRG.

17- The last paragraph of Sec. 1 is hard to follow. List the questions first, then give an actual outline of the rest of the paper.

18- At the end of Sec. 4, what do you mean by "accurate numerics"? ED? DMRG?

Recommendation

Ask for major revision

  • validity: good
  • significance: high
  • originality: good
  • clarity: good
  • formatting: good
  • grammar: good

Author:  Christoph Karrasch  on 2025-09-23  [id 5850]

(in reply to Report 1 on 2025-07-16)

We thank the referee for the detailed feedback of our work and the recommendation for publication under the condition of a major revision. We did our best in addressing the multitude of points raised in the review.

1- Is it known whether the predicted active space (i.e. HOMO, LUMO, and nearby shells) match experimental or ab-initio predictions? I mean multiplicities, point-group irreps, etc. This would be crucial to know in order to assess whether the Hubbard model is a good description of the fullerene molecules.

The normal approach in quantum chemistry is to treat the pi electrons as independent from the sigma electrons that form the three bonds to the neighbouring atoms. Then we have one electron per atom and the foundational description is the Hückel model, which is equivalent to the single-band tight-binding model of condensed matter. The Hückel energy levels already give the correct order of magnitude for the observed energy levels and gaps, see, e.g., Manini & Tosatti (https://arxiv.org/pdf/cond-mat/0602134). The Hubbard model is regarded as a reasonable starting point to include the effects of interactions. For example, it leads to good agreement with the experiment in predicting the DOS gap in C60 monolayers using QMC (https://arxiv.org/abs/cond-mat/0610823).

It is generally believed that including longer-ranged Coulomb interactions should produce better results, hence there is a number of papers that investigate the extended Hubbard model as well (which we mentioned and cited). But note: Had we already included longer-ranged interactions, referees would inevitably ask whether the model is too complicated and whether pair binding can already be captured by the simpler Hubbard model - and this is exactly the question that we want to answer in this paper.

Overall, it stands to reason that the minimal model that can capture the lifted degeneracy of the different spin states is the Hubbard model. For the study of the fullerenes it has a long history since the 1990s and we are not proposing anything new here. It makes sense to investigate further model refinements in subsequent publications rather than to jump to the most complex model right away.

We would also note that we have recently investigated the Haldane phase in chains of carbon nanoflakes (https://arxiv.org/abs/2412.08252). These are related molecular carbon systems, where the Hubbard model is also regarded as a good starting point. We found that including longer-ranged Coulomb interactions only led to minor quantitative improvements.

We have now added a brief discussion of the relevance of the Hubbard model in the revised manuscript.

2- Table 1 is quite thin on information. Some useful additions would be: - the irrep formed by the relevant HOMO/LUMO in addition to the degeneracy - for the open shells, state how many electrons would fit in them in addition to how many they contain - is r the Hamiltonian range in real space or in the hybrid real/momentum space introduced in Sec. 2.1? How does it change upon the basis transformation? - for C60 at least, a figure detailing the layout of the active space (i.e. the energy, multiplicity and point-group irreps of all shells near the Fermi energy) would be desirable

We have added the irrep to the table. The shell capacity can be read off from the degeneracy. The range r is in real space and in our experience serves as a proxy for the complexity of the model in real space. After the basis transformation defining a range becomes less useful, since the whole Hamiltonian including the interaction term becomes long-ranged. We have added Fig. 2, which lays out all of the irreps for all shells for the molecules considered in the paper.

3- Section 2.1: The implementation of the ZN symmetry is the most interesting technical aspect of the paper, but very little detail is provided. In particular: - How do you order the orbitals cik for DMRG? Are the sites that make them up automatically sorted together by the real-space Cuthill-McKee algorithm, or do you need to label them manually? - How do you implement Eq. (4) and the hopping term? To someone who's not an expert in this technique, this looks like a daunting task. You should give a detailed discussion of how to obtain the Hamiltonian in MPO (?) form, at least in the simple charge-conserved case. - The first sentence of the last paragraph is quite confusing. Do you mean that the eigenvalue under the ZN rotation can be imposed as a symmetry quantum number, similar to charge?

In the ℤ_N case, we do not employ Cuthill-McKee at all, but rather the analogous procedure to treating the Hubbard model on a cylinder: The sites that are related by the ℤ_N rotations form cycles of length N, and we enumerate them consecutively within each cycle, and then move on to the next-closest cycle. The hopping matrix is block-diagonal so far. Additional off-diagonal hopping elements couple the different blocks/cycles. By doing the basis transformation block-wise, the long-rangedness of the interaction terms is then limited to the sites within each cycle.

The implementation of the transformed Hubbard Hamiltonian is indeed quite involved, but the basic idea is this: Each one-body term cdagc or two-body term cdagcdagcc is a simple product of operators that has a trivial MPO representation of bond dimension 1. We also know how to sum MPOs, but the bond dimension of the sum will blow up. The task thus boils down to an MPO compression problem that minimizes the MPO bond dimension, but adds no losses. For this, we have implemented the algorithm by Hubig, McCulloch & Schollwöck (2017), which works quite well in our experience. For the spin-SU(2)-invariant case, some more work is involved. We first regroup the terms according to the SU(2)-invariant processes like spin exchange, pair hopping, and so on. How these terms map to the SU(2) operators used internally in the code is well-documented in the paper by Keller and Reiher (2016).

We have now written a more detailed explanation of the ℤ_N procedure, but since it has become quite lengthy, we have moved it to App. B along with the previous Fig. 2 (now Fig. 10), which illustrates the process for C20.

4- Report the energies and variances of all DMRG runs plotted in Figures 3-9, either as tables in the appendix or as supplementary text files. These would be useful both for benchmarking and to independent assessment of the extrapolated results and conclusions.

We have done so in the revised manuscript, App. C.

5- Figures 3-9: include error bars (i.e. the standard intercept error of the linear fit) for all extrapolations, as they are essential to assess the confidence one should put into the results. Also make sure they're visible (in Fig. 8, the only figure that has it, it's easy to miss, because it's just a vertical line overlapping with the grid line, adding the usual serifs at the end would help). If the error bars are too small to see, state that in the caption explicitly.

We have added error bars with serifs to the figures as requested in the revised manuscript.

6- Figures 3-9: The aspect ratio of the figures makes it look like there's almost no extrapolation, and it's also hard to see how accurately the points fit to the extrapolation lines. Consider laying out the panels horizontally, so that each of them are more square-like.

We have changed the layouts of the relevant figures accordingly in the revised manuscript.

7- Figure 9: It's difficult to see how good the straight line fit is given all data points are crammed in one corner. Add extra panels/insets zooming in on the data points.

We have added an inset that zooms in on the datapoints.

8- Table 2 (and error bars of Figure 9): - Show the standard intercept error of the linear fit instead of the custom error estimate. (If you want to use the separate data sets, show the full intercept+error pair for both.) - Comment on the fact that the extrapolated ground-state energies for 62 and 63 electrons are equal to within 1σ (62 electrons: Δ=0.009, σ=0.008,0.003, 63 electrons: Δ=0.004, σ=0.004,0.001). How much should we still trust the ordering of the energy estimates?

We have now replaced our previous error estimate by the standard error (of the extrapolated intercept).

Regarding the overlap within 1σ: It is clear that the C60 data should come with a disclaimer. One gets used to the numerical exactness of DMRG for 1D systems, but for this problem, one has to treat it as an approximative method. It is, however, still a controlled method. The relevant parameter is the energy variance per site, which is implementation-independent, and allows for a comparison with other, potentially better methods in the future.

We have added corresponding remarks to the manuscript.

9- Do the conclusions from DMRG match predictions from simpler calculations (e.g. ED restricted to the HOMO/LUMO)?

We have tested this and find that ED restricted to the HOMO & LUMO predicts a maximum-spin ground state. However, this method is uncontrolled. We have tested the same for C12 where full ED gives a minimum-spin ground state, but we find that ED restricted to the HOMO & LUMO predicts a maximum-spin ground state again.

In fact, we have observed the same in nuclear physics, where we find that taking more shells into account than is customary still quantitatively changes the results (https://onlinelibrary.wiley.com/doi/pdfdirect/10.1002/andp.202300436). In momentum space, the energy levels are coupled by many terms which seem to be only negligible for levels with very high energy separation.

10- Is the two-stage breakdown of Hund's rule in C28 related to the accidental degeneracy of the T2 and A1 irreps? What would Hund's rule predict if the degeneracy were lifted?

Regardless of how the degeneracy is lifted, from a simple Hückel + Hund's rule consideration, we expect an S=1 ground state in this case and then a one-stage breakdown similar to C20.

11- Z(N) is nonstandard notation for the group in question. The typical mathematical notation is ℤ_N, but given the geometrical meaning of the symmetry group, CN is even more appropriate. Switch to either one of these.

We have changed the notation to ℤ_N and mentioned that it is equivalent to C_N.

12- In the abstract and the introduction: "massive, large-scale" is a bit excessive, "large-scale" would be enough.

We have amended the text accordingly.

13- In the abstract: "For C40, Hund's rule is broken in the singlet ground state" Do you mean "at half filling"?

Yes, we have amended the text accordingly.

14- In the first paragraph of the introduction: Is the discussion of the nuclear anti-Hund's rule really relevant? If so, provide reference(s).

We find that most condensed-matter physicists are often unaware of the connection with nuclear physics and would like to point that out in order for the communities to learn from each other. The pair binding energy we consider here is for example known as the "even-odd mass difference" and relates different isotopes to each other (where the shells are filled by neutrons rather than electrons).

We have now added references to this introductory statement.

15- Eq. (1): What is Ntot? How should one set it?

The question is not entirely clear to us. N_{tot} is the particle number as stated in the text. It is well-defined for a finite system.

16- Figure 1: what do the colours of the plaquettes mean? Also, show the numbering of the sites used for DMRG.

The colours merely highlight the hexagonal vs. pentagonal faces and show how the frustrated pentagons are completely separated from each other for C12 and C60, and we qualify these systems as "weakly frustrated" (cf. the discussion in App. A).

We have added the site numberings to the new Fig. 1.

17- The last paragraph of Sec. 1 is hard to follow. List the questions first, then give an actual outline of the rest of the paper.

The paragraph in question is a review of the current state of research and knowledge about the problem. This discussion is necessary to formulate the questions of the paper in the first place. We have now shortened it, added an outline of the paper, and moved some of the finer details regarding C20 and C60 to the respective sections.

18- At the end of Sec. 4, what do you mean by "accurate numerics"? ED? DMRG?

Both. When spatial symmetries are exploited, the Heisenberg model on C28 is solvable by ED. We have no access to such a sophisticated code, but can also accurately solve the model with our DMRG code. The formulation in the paper has been clarified.

---

## Round 1 · Referee Report · Anonymous (Referee 2) · 2025-7-23

Strengths

Technically very advanced density matrix renormalization group simulations

Weaknesses

Only ground-state energies and differences are computed
No discussion of possible correlations in the system
limited discussion of the implications of the results

Report

The authors study the Hubbard model in fullerene geometries up to size 60 using the density matrix renormalization group method. Thereby, it is revealed that, depending on the interaction strength and the precise cluster, the ground state is a spin singlet, triplet, or quintuplet, and pair binding energies relevant for Cooper pairing are analysed. The technical skills of the authors using DMRG are very impressive. Combining SU(2) spin symmetry together with Z(N) is an admirable technical achievement, and it can be safely assumed that the simulations are performed in a competent way. This is also verified by comparison to exact diagonalization on smaller geometries.

I think the data is well presented, the main message is clear, and the paper is easy to read. However, my main criticism is that only relatively simple ground-state energies and derived quantities are compared, and no discussion of any correlations on these clusters is performed, which would be required to establish a picture of the physics in these compounds. As such, I am not sure enough new insights are presented to warrant publication in SciPost Physics. Of course, the data presented appears reliable and definitely worth publishing. I would instead suggest that the results are published in SciPost Physics Core, if the authors do not wish to extend their findings beyond ground state energies.

I do have several minor comments also, which the authors should incorporate before resubmitting.

1) While the abbreviation LUMO is introduced, the abbreviation HOMO is not introduced. This should be added.

2) The authors state the pair binding energies in absolute numbers, but do not discuss the implications. What would help is a comparison to similar pair binding energies reported in verified superconducting phases of the Hubbard model, to give an understanding of how strong the pairing is.

3) At the end of page two the authors write: "we give the best possible estimate using massive, large-scale DMRG computations that are significantly more advanced than anything that can be found in literature". Like this (and even given the surrounding context) it sounds like the authors want to claim that their code is currently the best DMRG code available. Such a claim would need to be backed by actual benchmarks to a wide variety of distinct codes, and ideally open-source accessibility of the code. Since I doubt the authors intend to do so, I suggest lowering this statement to something like, "state-of-the art DMRG code with which we produce the best current estimates for the given problem". Anyone familiar with DMRG is already impressed with their technical achievements, so no need to have such a sentence.

4) Figures 3,5,7,9 show extrapolations of ground state energies for different geometries with different spin and particle number. The issue is that the aspect ratio chosen in these plots is a bit fishy, since all the extrapolations look good like this, since they all lie on a flat line. The authors should make this aspect ratio to be close to 1:1, such that the quality of the extrapolation can be better assessed. I would suggest instead of having three rows of panels, to plot three columns of panels.

5) On page 7, the extrapolated ground state energy from DMRG is E0=-20.59211 which is lower than the quoted ED result, E0=-20.59202. A comment why a variational energy is lower (I suspect imperfect extrapolation) than the exact ground state energy is in order.

6) I very much appreciate the efforts to have error estimates for the C60 cluster in Table 2. The manuscript could be improved if all newly computed energies were accompanied with similar estimates.

7) Finally, for a non-expert on fullerene physics, it would help to describe in the introduction why the Hubbard model on fullerenes is relevant. Is it an interesting scientific question by itself (that's fair), or is it an accurate model of processes in the C60 molecule? By reading the introduction, I felt this question was not answered.

Requested changes

Change aspect ratio of Fig. 3,5,7,9
Implement minor changes described above

If publication in SciPost Physics is desired, a discussion of correlations would definitely strengthen the results.

Recommendation

Accept in alternative Journal (see Report)

  • validity: top
  • significance: good
  • originality: good
  • clarity: high
  • formatting: excellent
  • grammar: excellent

Author:  Christoph Karrasch  on 2025-09-23  [id 5849]

(in reply to Report 2 on 2025-07-23)

We thank the referee for the positive evaluation of our work and the assessment that it is "technically very advanced". We would, however, argue that the requested computation of correlation functions is not really relevant to the central questions of the paper. Our arguments are given below. With this, we hope to convince the referee for a publication in SciPost proper.

1) While the abbreviation LUMO is introduced, the abbreviation HOMO is not introduced. This should be added.

We have added the explanation of HOMO at the first occurrence.

2) The authors state the pair binding energies in absolute numbers, but do not discuss the implications. What would help is a comparison to similar pair binding energies reported in verified superconducting phases of the Hubbard model, to give an understanding of how strong the pairing is.

The standard conversion factor for carbon is t=2.7~2.8eV. The binding energies shown in the paper are all repulsive. In the paper by White & Kivelson, where attractive pair binding appears for smaller molecules, E_b is on the order of 10^{-2}t. This would be on the order of room temperature 0.025eV~300K. It is reasonable to compare it with the T_c of the C60 lattice, which is, however, an order of magnitude smaller. Thus we realistically expect a binding energy on the order of 10^{-3}t. This illustrates the difficulty of the problem: One needs energies resolved to about 10^{-4}t-10^{-5}t to be able to compute it.

For this reason, we are only able to compute the pair binding energy for C20, where it turns out to be repulsive, so unfortunately no meaningful comparison with superconducting phases of the Hubbard model is possible.

3) At the end of page two the authors write: "we give the best possible estimate using massive, large-scale DMRG computations that are significantly more advanced than anything that can be found in literature". Like this (and even given the surrounding context) it sounds like the authors want to claim that their code is currently the best DMRG code available. Such a claim would need to be backed by actual benchmarks to a wide variety of distinct codes, and ideally open-source accessibility of the code. Since I doubt the authors intend to do so, I suggest lowering this statement to something like, "state-of-the art DMRG code with which we produce the best current estimates for the given problem". Anyone familiar with DMRG is already impressed with their technical achievements, so no need to have such a sentence.

We have changed the formulation in the revised manuscript according to this suggestion.

4) Figures 3,5,7,9 show extrapolations of ground state energies for different geometries with different spin and particle number. The issue is that the aspect ratio chosen in these plots is a bit fishy, since all the extrapolations look good like this, since they all lie on a flat line. The authors should make this aspect ratio to be close to 1:1, such that the quality of the extrapolation can be better assessed. I would suggest instead of having three rows of panels, to plot three columns of panels.

This overlaps with the critique of referee 1, and we provide a 1:1 aspect ratio in the revised manuscript.

5) On page 7, the extrapolated ground state energy from DMRG is E0=-20.59211 which is lower than the quoted ED result, E0=-20.59202. A comment why a variational energy is lower (I suspect imperfect extrapolation) than the exact ground state energy is in order.

Yes, this is indeed the extrapolation error. We now provide the full data in App. C, and it can be checked that the DMRG results are always above the ED results before the extrapolation.

We note that the extrapolation is still useful, as it can provide us with an additional correct digit, see e.g. C20, Ntot=21, S=1/2: DMRG best: -19.6323 DMRG extrapolated: -19.6334 ED: -19.6332

6) I very much appreciate the efforts to have error estimates for the C60 cluster in Table 2. The manuscript could be improved if all newly computed energies were accompanied with similar estimates.

We have now added error bars from the linear fits to all figures, as well as all the datapoints in numerical form in App. C.

7) Finally, for a non-expert on fullerene physics, it would help to describe in the introduction why the Hubbard model on fullerenes is relevant. Is it an interesting scientific question by itself (that's fair), or is it an accurate model of processes in the C60 molecule? By reading the introduction, I felt this question was not answered.

This overlaps with the reply to referee 1, please see our reply there. We are not the first ones who propose studying the Hubbard model on C60, this was done at least since the 1990s. The Hubbard model is the minimal model that lifts shell degeneracy and allows for a minimal/maximal-spin ground state. It was used previously to successfully compare photoemission spectra to the experiment. The Hubbard model is also widely used for coupled carbon nanoflakes to great success (see e.g. https://arxiv.org/pdf/2307.00991).

In this paper, we are addressing a gap in knowledge whether the Hubbard model predicts a minimum-spin state for high-symmetry fullerenes. We absolutely agree that the results can be improved in the future by considering more sophisticated models.

Finally, we note that the Hubbard model is already more realistic than other models that were considered for C60, such as the t-J or the Heisenberg model (unrealistic because we are not at strong coupling).

If publication in SciPost Physics is desired, a discussion of correlations would definitely strengthen the results.

This is not a strictly requested change, but we want to comment anyway: In investigating condensed-matter systems, we are of course interested in correlation functions. For superconductivity, we would consider long-ranged pairing correlations. For the fullerides, this would require an actual lattice of C60 (monolayer or fcc). Since this is a hopelessly intractable problem, we focus on the proxy quantity of the pair binding energy and the spin of the ground state of a single molecule (which is complicated enough). Computing correlation functions within a single molecule makes no sense here, as it doesn't advance the main question of superconductivity in the fullerides. We could still investigate the nature of the Hubbard ground state on a single C60 as a problem in its own right (one of the authors did so for the Heisenberg model: https://www.scipost.org/10.21468/SciPostPhys.10.4.087), but we feel that it would only distract from the main question of paper. We think that our paper is already long enough. The meat of it is to go into the breadth of molecules, so that we present results for C20, C28, C40 beside C60, rather than going into the depth of understanding the C60 ground state.

---

## Round 2 · Referee Report · Anonymous (Referee 1) · 2025-10-14

Strengths

As before

Weaknesses

Insufficient accuracy to establish Hund's rule violation in the $C_{60}$ Hubbard model

Report

I thank the authors for considering and incorporating most of my requested changes into the revised manuscript. In particular, the approach the authors take to implement the rotational symmetry is explained much more clearly and the results are presented in easier-to-read graphs as well as detailed tables.

However, I am still not convinced about the apparently central scientific point, the violation of Hund's rule in electron-doped $C_{60}$. If clearly evidenced, this would serve as a proxy for an electronic pairing mechanism that may underlie superconductivity in alkali-fullerene compounds. This would've been quite a remarkable result and indeed, this appears to be the authors' "selling point" for publication in SciPost. However, as it's now clear in Fig. 9, the ordering of the different spin sectors is quite ambiguous for 63 and 64 electrons, and it's unlikely this would improve without much larger-scale (or otherwise improved) simulations.

I'm also unhappy that the authors paid little attention to the other referee's suggestion to consider correlation functions. While DMRG energies may be hard to push beyond what's in the paper, correlation functions could've given the authors an alternative way to find more solid evidence for electronic pairing and enhance the significance of their paper.

The technical aspects of the paper are still quite impressive, but I do not think they warrant publication in SciPost Physics on their own. Without an unambiguous physical result with potential relevance for fullerene superconductors, I believe the paper is more suitable for SciPost Physics Core.

Requested changes

1- Re my earlier question on Eq. (1): $E_b$ obviously depends on $N_{tot}$ and in any given compound, there should be a particular value that is relevant. It's not always clear what this value is in the graphs and otherwise. For example, what is it in Fig. 4? Based on Fig. 3, I'd guess it's $E_{20}+E_{22}-2E_{21}$, which doesn't sound like the "half-filled ground state". Wouldn't $E_{19}+E_{21}-2E_{20}$ make more sense? This should be clarified in the text for $C_{20}$ and maybe comment on what $N_{tot}$ is really relevant for $C_{60}$.

2- In Fig. 9, middle panel, the circles around the extrapolated values cover the serifs of the error bars. A symbol that marks the extrapolated value itself (like a cross or a third serif in the middle) might be more useful for the other plots too?

Recommendation

Accept in alternative Journal (see Report)

  • validity: good
  • significance: good
  • originality: good
  • clarity: high
  • formatting: good
  • grammar: good

Author:  Roman Rausch  on 2025-10-17  [id 5941]

(in reply to Report 1 on 2025-10-14)
Category:
remark
reply to objection

Comment:
We did address the request for correlation functions, but the referees didn't counter-address it. Long-range pairing correlations *on the molecular lattice* would provide evidence for electronic pairing, but this problem is intractable. The energies of a single molecule serve as proxy quantity for pairing on the lattice (as we explain in the introduction), but we can't think of a intra-molecular correlation function that would enhance this evidence; and there has been no explicit suggestion by the referees.
Furthermore, energies generally converge faster than correlation functions in DMRG, and so provide the highest-quality data.
Finally, this request would require repeating numerical calculations that take weeks to months to complete.

---

## Round 2 · Referee Report · Anonymous (Referee 2) · 2025-10-15

Report

The authors have implemented the requested changes to my previous report. In particular, the scaling of ground state energies with the new aspect ratio makes the extrapolations more credible. I remain with my assessment that the study is solid, the numerical technology is state-of-the-art, and it will be of interest to several physicists, so it should be published without any further modifications. However, even after the resubmission, I do not see strong enough reasons justifying publication in SciPost Physics, but would rather suggest SciPost Physics Core as a suitable venue for publication.

Recommendation

Accept in alternative Journal (see Report)

---

## Round 2 · Author Response

Dear Editor,

We hereby resubmit our manuscript with a major revision that addresses the comments of the referees.

We see that the referees basically give no objections regarding our results, but are mostly curious about the validity of the Hubbard model for the fullerenes. But this seems to miss the point: The aim of the paper is not to argue for the relevance of the Hubbard model for fullerenes, but rather to fill a hole in the knowledge of whether or not it predicts pair binding. The Hubbard model has been studied for fullerenes since the 1990s and there is consensus that it is a valid starting point. In the revised manuscript, we give a brief discussion of the past works, in particular regarding the agreement with the experiment in terms of single-particle spectra. Of course, more complex models (e.g., with long-ranged Coulomb interaction) are expected to have better agreement, but this is always the situation with model systems; and such complexity is best build one step at a time.

We have adjusted the figures and tables in accordance with the requests of the referees and provide the numerical energy values in the appendix. With these changes, we hope for an acceptance of our paper in SciPost.

Referee 2 proposes a downgrading to SciPost Core because correlation functions were not computed, but we argue that this might also miss the point: Correlation functions are irrelevant for the questions we are addressing here. We are after the proof of pair binding, we are not after a deep analysis of the nature of the ground state for the molecules (which in any case will only be true within the Hubbard model).

Best Regards,
Roman Rausch and Christoph Karrasch

---

## Round 2 · List of Changes

• added the irrep to Tab. 1
  • added Fig. 2.
  • added error bars to the figures
  • optical changes to the figures (e.g., aspect ratio)
  • expanded on the DMRG method (now App. B)
  • documented raw data (App. C)
  • added a discussion of the relevance of the Hubbard model
  • changes to the wording (see replies to the referees)

---

## Round 3 · Author Response

We made minor modifications in order to implement the suggestions of Referee 1.

---

## Round 3 · List of Changes

We clarified the meaning of Eb and used different symbols in the figures to improve the optics.

---

## Editorial Decision

accepted_in_target_journal